# Exploring the Valorization of Buckwheat Waste: A Two-Stage Thermo-Chemical Process for the Production of Saccharides and Biochar

Yongheng Yuan [1], Faqinwei Li [1], Nanding Han [2], Bingyao Zeng [1], Yoshiaki Imaizumi [1], Risu Na [1] and Naoto Shimizu [2,*]

[1] Laboratory of Agricultural Biosystem Engineering, Graduate School of Agriculture, Hokkaido University, Sapporo 060-8589, Japan
[2] Research Faculty of Agriculture/Field Science Center for Northern Biosphere, Hokkaido University, Sapporo 060-8589, Japan
* Correspondence: shimizu@bpe.agr.hokudai.ac.jp or shimizu@eis.hokudai.ac.jp; Tel.: +81-117063848

**Abstract:** To realize the utilization of the valorization of buckwheat waste (BW), a two-stage thermal-chemical process was explored and evaluated to produce saccharides and biochar. During the first stage, BW underwent a hydrothermal extraction (HTE) of varying severity to explore the feasibility of saccharides production; then, the sum of saccharides yields in the liquid sample were compared. A higher sum of saccharides yields of 4.10% was obtained at a relatively lower severity factor (SF) of 3.24 with a byproducts yield of 1.92 %. During the second stage, the contents of cellulose, hemicellulose, and lignin were analyzed in the residue after HTE. Enzymatic hydrolysis from the residue of HTE was inhibited. Thus, enzymatic hydrolysis for saccharides is not suitable for utilizing the residue after HTE of BW. These residues with an SF of 3.24 were treated by pyrolysis to produce biochar, providing a higher biochar yield of 34.45 % and a higher adsorption ability (based on methyl orange) of 31.11 % compared with pyrolysis of the raw BW. Meanwhile, the surface morphology and biomass conversion were analyzed in this study. These results demonstrate that the two-stage thermal-chemical process is efficient for treating BW and producing saccharides and biochar. This work lays a foundation for the industrial application of BW, and for improving the economic benefits of buckwheat cultivation.

**Keywords:** buckwheat waste; two-stage thermal-chemical process; saccharides and biochar; enzymatic hydrolysis inhibition





## 1. Introduction

Lignocellulose biomass energy plays a vital role in the whole energy supply system, as the fourth largest energy resource after fossil fuels. It has been widely utilized due to its low value, carbon-neutrality, and extensive availability [1]. One important aspect of lignocellulose biomass is agricultural waste, such as crop straws [2,3].

Buckwheat is widely cultivated around the world, with a production of more than 2.90 million tons in 2018 [4]. It has been planted for food and medicine, due to the amino-acid composition and its contents of fiber, resistant starch, trace elements, vitamins, and antioxidants [5–7]. Buckwheat waste (BW), which includes husks, leaves, and straw, is usually thrown away or burned. This leads to serious pollution problems and a waste of resources. BW is rich in cellulose, hemicellulose, antioxidants, and various sugars [4,8,9]; thus, it can be considered as a potential source for improving the economic benefits of buckwheat cultivation. Kim et al. reported that phenolic compounds can be produced or extracted from buckwheat leaves using subcritical water at different temperatures and holding times [10]. Kraujalienė et al. evaluated the recovery of phytochemicals from buckwheat flowers using supercritical liquid extraction [11]. Activated carbon and biochar from buckwheat hulls were prepared by Pena et al. for the catalytic purification of

syngas [12]. Elsayed et al. searched the effect of the methane potential of buckwheat husks and straw in the inoculum volatile solids, inoculum to substrate ratio, and other factors [13].

However, many problems remain. First, most studies have focused on either the husks, leaves, or straw of buckwheat. In industrial applications, separation of these components from one other is tedious work. Second, BW contains many antioxidants as well as cellulose, glucose, and xylose, while few studies explored the saccharides produced from BW. Finally, residue after extraction is usually discarded, which is a waste of resources and causes contamination from the added chemicals. To overcome these three problems, a two-stage process was considered involving the production of saccharides from BW (mixture of husk, straw, and leaves), then conversion of the residues into adsorbent materials (biochar) through appropriate treatment methods.

Several treatment methods including thermal-chemical treatment (pyrolysis, liquefaction, and gasification) have been developed to improve the desired products yield [14–17]. Among them, the hydrothermal method is popular since it can break down the chemical bonds in lignocellulose using high temperatures and pressures, which hydrolyze the cellulose and remove the hemicellulose and some lignin without added chemicals [15]. Research interest in pyrolysis has increased because no dioxins are formed during the process, and it uses milder temperatures compared with those used in gasification [18,19].

In current study, the thermal-chemical process was combined with the problems in the BW utilization. A relatively mild hydrothermal extraction (HTE) was chosen as the first stage for cellulose and hemicellulose degradation to produce saccharides from raw BW. Then, in the second stage, the stubborn solid residue which could not be degraded during the first stage of HTE was used for biochar production by pyrolysis. The surface morphology of biochar is observed by SEM to illustrate the mechanism of adsorption. Moreover, the HTE of BW resulting in the inhibition of enzymatic hydrolysis was illustrated. Therefore, the objective of this study was to provide valuable information on the utilization of BW as a renewable resource in the biorefinery industry for the production of saccharides and biochar.

## 2. Materials and Methods

### 2.1. Materials

BW (from the Kitawase Soba variety) was obtained from the Hokkaido University test field, Sapporo, Japan, in September 2021. The dried BW was ground in a mill, then sieved by a 40-mesh screen. The powder of BW was dried at 25 °C until it reached a constant weight and was then sealed in plastic bags before storage in a drying oven at room temperature before use. A schematic diagram of the study is presented Figure 1.

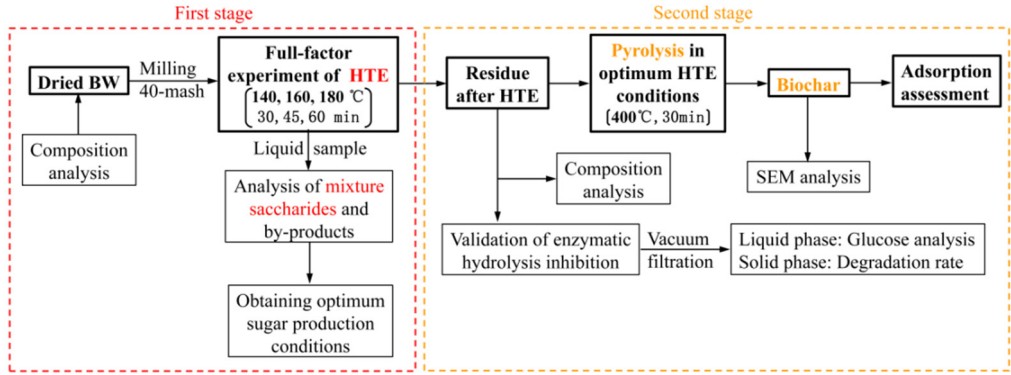

**Figure 1.** A schematic diagram for the two-stage thermal chemical process of BW.

### 2.2. HTE Experiment

HTE was performed in a 150-mL stainless-steel batch reactor (PPV-3461 Chemi-Station, EYELA, Tokyo, Japan) equipped with a heating jacket, an embedded thermometer, a magnetic agitator, and a cold-water circulation system (NCB-2500, EYELA, Tokyo, Japan).

A 10.0-g sample of the raw BW was mixed with 100 mL of distilled water and was then kept at 25 °C (control groups) or heated at 140, 160, and 180 °C and held at each temperature for 30, 45, and 60 min. The mixtures (BW with distilled water) were heated from an initial temperature of 23 ± 1 °C to the target temperature at 4 °C/min while being agitated at 500 rpm. The holding time was measured from when the target temperature was attained. The pressure within the reactor was achieved solely through the rise in temperature during the treatment. After reaction, the reactor was taken out from the heating jacket and cooled to room temperature (25 °C) by the circulation of cold water at 5 °C so that the blended solid-liquid mixture could be removed. The reactor and the sample were washed by distilled water, and the solids were separated by vacuum filtration and dried in an oven at 50 °C to a constant weight before analysis. All experiments were repeated in triplicate.

To compare the comprehensive effects of temperature and holding time during the HTE, a severity factor (SF) defined as $\log R_0$ was introduced using the following equation [20]:

$$R_0 = t \times \exp\left[\frac{T - 100}{14.75}\right] \tag{1}$$

where T is the reaction temperature (°C) and t is the holding time (min).

*2.3. Analysis of Hydrolysis Products*

The compositions of saccharides (glucan, xylan, cellobiose, glucose, xylose, and arabinose) and the byproducts (acetic acid, lactic acid, and 5HMF) in the hydrolysate were determined by HPLC with an RI detector (Agilent 1260 series, Agilent Technologies, Santa Clara, CA, USA) and columns for saccharides analysis: KS-802 and guard-column, KS-G 6B (Shodex). The column temperature was set at 50 °C with Milli-Q-grade water used as the mobile phase at a flow rate of 0.7 mL/min. For the byproducts analysis, the columns were SH-1821 and guard-column SH-G (Shodex). The column temperature was set at 60 °C with a 2-mmol $H_2SO_4$ solution as the mobile phase at a flow rate of 0.6 mL/min. The pH of the hydrolysate was determined using a WD-35634-30 digital pH meter (Oakton Instruments, Vernon Hills, IL, USA). The 3,5-dinitrosalicylic acid (DNSA) method was used to determine the reducing sugars content [21]. The pH meter was validated by 4.01 (phthalate), 6.86 (phosphate), and 9.18 (tetraborate) of pH standard solution.

To consider the balance between the sum of the saccharides yield (SSY) and the energy consumption (expressed in SF), the SEC was used as an index to compare the results of the HTE experiments. The SEC was calculated as follows:

$$SEC = \frac{SSY\ (\%)}{SF} \tag{2}$$

where SEC is the specific energy consumption of HTE for saccharides production. SSY is the sum of the yields of cellobiose, glucose, xylose, and arabinose.

*2.4. Analysis of Lignocellulose Composition in Residue*

The lignocellulose composition of the residues, including the contents of total solids (TS), cellulose, hemicellulose, and lignin (acid-soluble lignin and acid-insoluble lignin) was determined in accordance with the NREL protocol [22,23].

The percentage solid recovery and the percentage removal of cellulose, hemicellulose, and lignin were calculated as follows:

$$Solid\ Recovery\ (\%) = \frac{Solid\ after\ treatment\ (g)}{Solid\ before\ treatment\ (g)} \times 100 \tag{3}$$

$$R_{(C,H,L)}\ (\%) = \frac{m_{(C,H,L)}\ before\ treatment\ (g) - m_{(C,H,L)}\ after\ treatment\ (g)}{m_{(C,H,L)}\ before\ treatment\ (g)} \times 100 \tag{4}$$

where $R_{(C, H, L)}$ is the percentage removal of cellulose, hemicellulose, and lignin, respectively, and $m_{(C, H, L)}$ is the mass of cellulose, hemicellulose, and lignin, respectively.

### 2.5. Validation of the Enzymatic Hydrolysis Inhibition of the Residue after HTE

An enzymatic hydrolysis experiment on the residue after HTE and the control groups was performed to explore the inhibitory effects. A sample of residue (0.5 g) was added to 25 mL of 0.1 M sodium citrate buffer (pH = 4.8) which included 0.5 mL of a 2% sodium azide solution to prevent microbial contamination in a 50-mL vial. Cellulase (MP Biomedicals, LLC, Santa Ana, CA, USA) was added to the mixture at 700 U/g substrate. Before the experiment, all utensils were sterilized in an autoclave at 121 °C for 10 min. The mixture was heated at 50 °C in a shock incubator at 180 rpm for 48 h. After 6, 12, 24, 36, and 48 h, samples of the liquid were collected and centrifuged before glucose analysis by HPLC. After 48 h of hydrolysis, the residue was collected by vacuum filtration and dried at 50 °C until to a constant weight to calculate the rate of degradation of the substrate. For glucose analysis by HPLC, an SH-1821 column and an SH-G guard-column (Shodex, Tokyo, Japan) were used. The column temperature was set at 60 °C with 2 mmol $H_2SO_4$ solution used as the mobile phase at a flow rate of 0.6 mL/min. All of the experiments were repeated in triplicate.

The percentage degradation of the substrate after 48 h of enzymatic hydrolysis was calculated by the following equation:

$$\text{Degradation of substrate } (\%) = \frac{\text{Substrate before hydrolysis (g)} - \text{Solid phase after hydrolysis (g)}}{\text{Substrate before hydrolysis (g)}} \times 100 \quad (5)$$

### 2.6. Pyrolysis Treatment to Produce Biochar

The pyrolysis experiment was conducted in a muffle furnace (FO300, Yamato Scientific, Tokyo, Japan). A 1-g sample of raw BW or the residue after HTE at 160 °C for 30 min (SF = 3.24) was incubated and heated in a 10-mL crucible at 400 °C for 30 min to produce the biochar. Nitrogen was introduced for the first 5 min to completely purge any air. The crucible was closed quickly, then the temperature of the muffle furnace was increased at 10 °C/min. The holding time was measured from when the target temperature was reached. After the reaction, the crucible was cooled down. All of the experiments were repeated in triplicate. The production rate was calculated as follows:

$$\text{Biochar production rate } (\%) = \frac{\text{mass of biochar (g)}}{\text{mass of residue after HT (g)}} \times 100 \quad (6)$$

### 2.7. Methyl Orange Adsorption Experiment

Methyl orange was used as a model compound to assess the adsorption capacity of the biochar from the residue after HTE [24]. A sample of biochar (0.3 g) was added to 30 mL of 8 mg/L of methyl orange solution in a 50-mL vial. Adsorption experiments were performed in a rotating shaker at 180 rpm for 7 h. Samples of liquid were collected after 0.5, 1, 2, 3, 5, and 7 h, filtered through a 0.45-um membrane, then their absorbance at 464 nm was detected by a spectrophotometer (V-560 UV/VIS, Jasco, Tokyo, Japan) [24]. The concentration of methyl orange was expressed according to a standard curve (concentration vs. absorbance). Triplicate experiments were conducted under the same conditions.

### 2.8. The Surface Morphology of Residue and Biochar

The surface morphology analysis of the biochar and control sample was performed with scanning electron microscopy (SEM) (JSM-6301F, JEOL Ltd., Tokyo, Japan) at 10 kV. The samples were sputter-coated with a thin layer of gold. Images were obtained at 6000 magnifications.

### 2.9. Statistical Analysis

The experimental data were presented as the means ± standard error (SE) of three replicates (*n* = 3). The ANOVA was performed using SPSS Statistical Software Version 25.0 to determine the statistical significance at 95% confidence interval. All the figures were prepared using OriginPro 2021 software (OriginLab, Northampton, MA, USA).

## 3. Results and Discussion

### 3.1. First Stage

Liquid Fractions after HTE of BW

In the present study, the pH decreased from 5.24 ± 0.03 to 4.23 ± 0.05 as the SF increased (Table 1). This could have been caused by water generating hydrogen ions under HTE due to autoionization [25] and the volatile fatty acids released or produced during HTE [26]. The yield of reducing sugars increased as the SF increased during HTE. The highest reducing sugars yield was 10.81 ± 0.32% at 180 °C for 60 min (SF = 4.45), 2.8 times greater than the yield from the control group (25 °C for 60 min). Figure 2 shows the linear relationships between SF and the pH, reducing sugars yield and percentage solid recovery. The pH of the liquid sample and the percentage of solid recovery decreased as the SF increased. The study of Batista et al. reported similar results: the SF of the sugarcane straw showed a negative correlation with mass yield and pH value after hydrothermal treatment [27]. The correlation between reducing sugars yield and SF was similar to that reported for the yield of reducing sugars from corn stover after hydrothermal pretreatment [26]. These results showed that increasing the SF can enhance the degradation of BW to produce reducing sugar, but will also produce some undesired byproducts (acids) at the same time.

**Table 1.** The chemical component, yield, and pH of the liquid sample in the first stage of different HTE conditions. Average ± s.d. Values in a column sharing an alphabet are not significantly different (*p* < 0.05).

| Sample | pH | RS[1] (*w/w*%) | (Oligo- and mono-) saccharides (*w/w*%) | | | | by-products (*w/w*%) | | |
|---|---|---|---|---|---|---|---|---|---|
| | | | Cellobiose | Glucose | Xylose | Arabinose | Acetic acid | Lactic acid | 5HMF[2] |
| $R_{30 min-25 °C}$ | 5.24±0.03 [a] | 4.03±0.06 [f] | 0.17±0.00 [e] | 1.95±0.20 [a] | 0.86±0.04 [d] | 0.10±0.02 [g] | 0.17±0.04 [e] | 0.18±0.01 [e] | 0.78±0.002 [d] |
| $R_{30min-140 °C}$ | 4.94±0.05 [b] | 4.82±0.31 [e] | 0.22±0.04 [de] | 1.42±0.07 [c] | 1.35±0.39 [c] | 0.14±0.02 [f] | 0.62±0.02 [d] | 0.25±0.02 [de] | 0.81±0.002 [d] |
| $R_{30min-160 °C}$ | 4.58±0.04 [d] | 6.25±0.30 [d] | 0.22±0.01 [de] | 1.81±0.01 [b] | 1.83±0.07 [a] | 0.25±0.01 [cd] | 0.83±0.07 [c] | 0.28±0.01 [de] | 0.81±0.016 [d] |
| $R_{30min-180 °C}$ | 4.39±0.11 [e] | 9.60±0.80 [b] | 0.45±0.03 [c] | 1.09±0.04 [d] | 1.49±0.01[abc] | 0.35±0.02 [a] | 1.64±0.30 [b] | 0.56±0.15 [c] | 0.94±0.070 [c] |
| $R_{45min-25 °C}$ | 5.25±0.02 [a] | 4.00±0.06 [f] | 0.16±0.02 [e] | 2.01±0.02 [a] | 0.85±0.03 [d] | 0.12±0.02 [fg] | 0.20±0.01 [e] | 0.25±0.03 [de] | 0.78±0.003 [d] |
| $R_{45min-140 °C}$ | 4.83±0.03 [c] | 5.23±0.15 [e] | 0.23±0.01 [de] | 1.53±0.02 [c] | 1.56±0.04 [bc] | 0.14±0.02 [f] | 0.73±0.04 [cd] | 0.27±0.02 [de] | 0.80±0.005 [d] |
| $R_{45min-160 °C}$ | 4.56±0.04 [d] | 6.82±0.21 [c] | 0.26±0.02 [d] | 1.77±0.06 [b] | 1.84±0.12 [a] | 0.31±0.01 [b] | 0.83±0.06 [c] | 0.30±0.05 [d] | 0.83±0.013 [d] |
| $R_{45min-180 °C}$ | 4.28±0.07 [f] | 10.60±0.30 [a] | 0.70±0.07 [b] | 1.09±0.03 [d] | 1.47±0.06 [c] | 0.27±0.00 [c] | 2.16±0.18 [a] | 0.74±0.01 [b] | 1.05±0.094 [b] |
| $R_{60min-25 °C}$ | 5.25±0.02 [a] | 3.92±0.20 [f] | 0.16±0.02 [e] | 2.01±0.04 [a] | 0.85±0.04 [d] | 0.13±0.02 [f] | 0.20±0.02 [e] | 0.24±0.05 [de] | 0.79±0.003 [d] |
| $R_{60min-140 °C}$ | 4.80±0.01 [c] | 5.29±0.19 [e] | 0.22±0.00 [de] | 1.53±0.01 [c] | 1.56±0.00 [bc] | 0.17±0.02 [e] | 0.79±0.01 [e] | 0.29±0.02 [de] | 0.80±0.004 [d] |
| $R_{60min-160 °C}$ | 4.55±0.05 [d] | 6.86±0.13 [c] | 0.27±0.02 [d] | 1.69±0.02 [b] | 1.74±0.06 [ab] | 0.32±0.00 [b] | 0.95±0.01 [e] | 0.33±0.07 [d] | 0.84±0.014 [d] |
| $R_{60min-180 °C}$ | 4.23±0.05 [f] | 10.81±0.32 [a] | 0.94±0.13 [a] | 1.10±0.05 [d] | 1.43±0.05 [c] | 0.23±0.01 [d] | 2.68±0.14 [e] | 1.01±0.08 [a] | 1.12±0.042 [a] |

[1] RS yield, Reducing sugar yield; [2] 5HMF, 5-hydroxymethylfurfural.

The components in the hydrolysate, such as oligosaccharides (cellobiose) and monosaccharides (glucose, xylose and arabinose), are given in Table 1. These saccharides were released from BW itself and produced from the degradation of cellulose and hemicellulose by HTE [28]. Compared with the control groups, the cellobiose yield increased after HTE, gradually from 0.22 ± 0.04% at 140 °C to 0.27 ± 0.02% at 160 °C, while there was no significant difference (*p* > 0.05). However, at 180 °C it clearly increased with the holding time with a significant difference (*p* < 0.05), reaching a maximum value of 0.94 ± 0.13% at 180 °C for 60 min (SF = 4.13). The yields of xylose and arabinose increased as the SF increased, then decreased as the SF further increased, with maximum yields of xylose and arabinose of 1.84 ± 0.12% and 0.35 ± 0.02%, respectively, obtained at 160 °C for 45 min (SF = 3.42) and 180 °C for 30 min (SF = 3.83). These results appeared because a large amount of cellulose and hemicellulose could be degraded into cellobiose and monosaccharides with the increase of SF of HTE [28], until the SF increased to 3.42 and 3.83. Then, some

xylose and arabinose were degraded to other byproducts in higher SF, resulting in a lower yield [28,29]. In contrast to the other monosaccharides, the glucose yield in the control groups was higher than that after HTE with significant difference ($p < 0.05$). The result may be because part of the glucose released from the BW itself was degraded during the HTE, while the amount of glucose produced by the degradation of cellulose and hemicellulose in the HTE was less than the degraded during HTE [28]. In the HTE groups, the tendency was similar to those of xylose and arabinose, with maximum yields of $1.81 \pm 0.01\%$, obtained at 160 °C for 30 min (SF = 3.24).

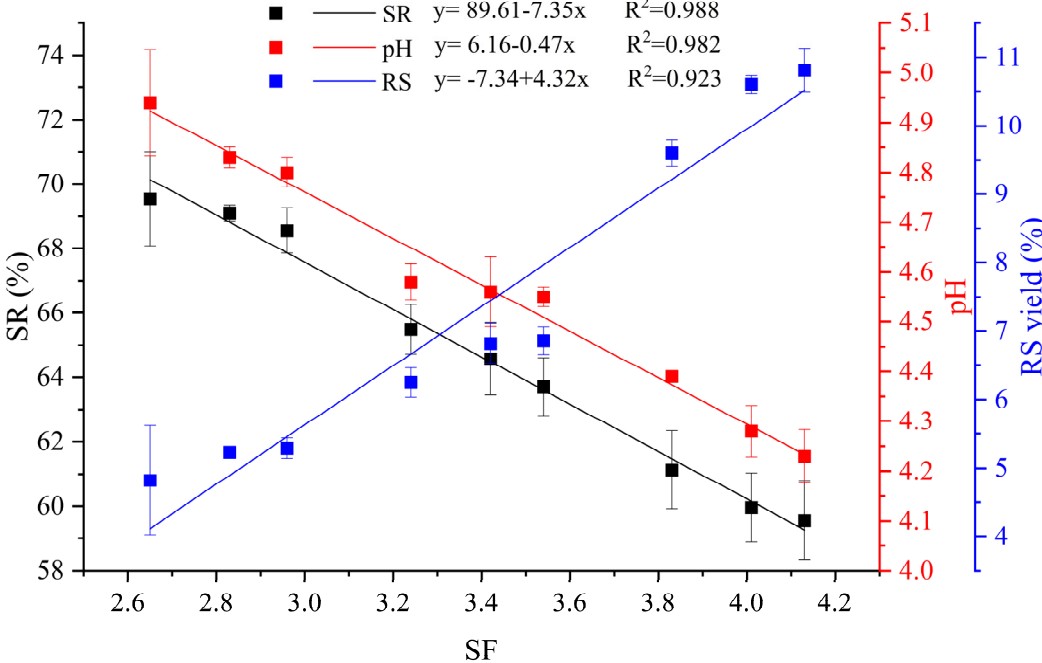

**Figure 2.** Relationships between the SF (severity factor), SR (percentage solid recovery), pH, and RS (reducing sugar) yield in the HTE of BW under the first stage.

Some byproducts in the hydrolysate (e.g. acetic acid, lactic acid, and 5HMF) were also detected, as shown in Table 1. These byproducts are released from BW and were produced from the degradation of cellulose, hemicellulose, and saccharides by HTE [28,29]. Compared with the control groups, only a small increase in the content of these byproducts was observed in mild HTE groups (SF = 2.65 to 3.54) ($p > 0.05$), while they dramatically increase ($p < 0.05$) with a reaction temperature of 180 °C. The maximum yields of acetic acid, lactic acid, and 5HMF were $2.68 \pm 0.14\%$, $1.01 \pm 0.08\%$, and $1.12 \pm 0.042\%$, respectively, all obtained under HTE conditions of 180 °C for 60 min (SF = 4.13). These indicated that the conditions before 160 °C led to a little degradation of saccharides, yet degradation of saccharides responded extremely at 180 °C.

The SSY (cellobiose, glucose, xylose, and arabinose) and SEC are shown in Figure 3. The higher SSY was obtained at 160 °C with a significant difference ($p < 0.05$) compared with other temperatures, while different residence time was not affected ($p > 0.05$). A maximum value of 4.18% was achieved at 160 °C for 45 min (SF = 3.42). It was less than our expected result because some of the polysaccharides in the hydrolyzate were not completely degraded to monosaccharides, as shown in Figure S1. The SEC was low at an SF of 3.83, 4.01, and 4.13, due to the degradation of saccharides to byproducts at the higher SF. Moreover, the SEC was much higher ($p < 0.05$) in the SF of 2.65, 3.24, and 3.42. The maximum SEC (1.27) was obtained at 160 °C for 30 min (SF = 3.24). In this case, a relatively larger saccharides yield (4.10%) was obtained with less energy consumption, while the byproducts yield (1.92%) was lower with a profitable percentage solid recovery (65.49% in Table 2), which could be reused in the second stage.

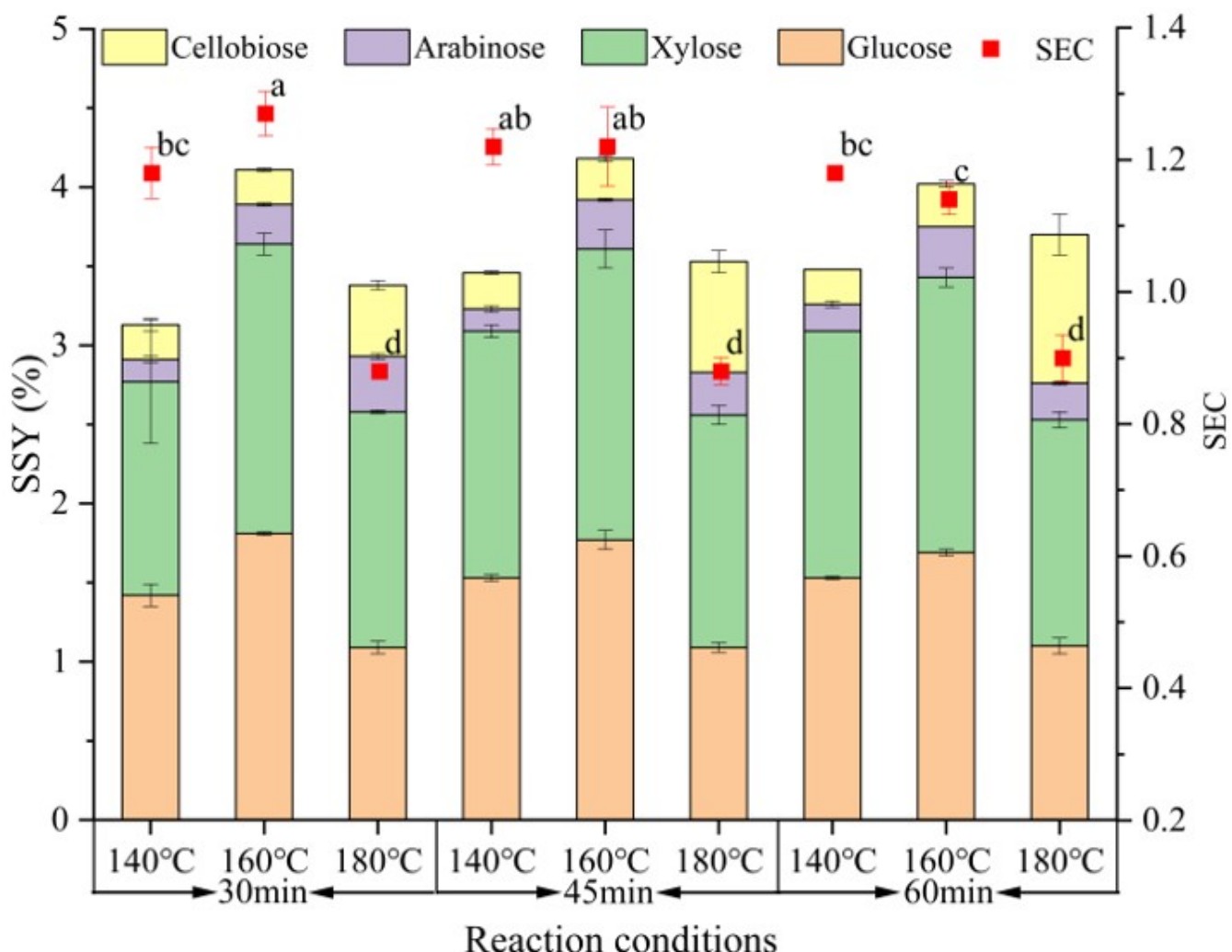

**Figure 3.** The sum of saccharides yield (SSY) in the first stage based on our detected component in the liquid sample and the SEC (specific energy consumption). The alphabets on the bar graph indicate statistical difference ($p < 0.05$).

**Table 2.** The percentage of solids recovery and chemical compositions of the residue of HTE and raw BW in second stage. Average ± s.d. Values in a column sharing an alphabet are not significantly different ($p < 0.05$).

| Samples | SF[1] (LogR_0) | SR[2] (w/w, %) | Chemical composition (w/w, %) | | | | | |
|---|---|---|---|---|---|---|---|---|
| | | | Cellulose | Hemicellulose | AIL[3] | ASL[4] | Total Lignin | TS[5] |
| Raw BW | | 100.00 | 27.77±0.93 [e] | 6.06±0.18 [bc] | 17.25±0.73 [d] | 8.87±0.73 [ab] | 26.12±1.06 [e] | 92.82±0.39 [f] |
| R_30min-25°C | | 86.75±0.85 [a] | 30.11±0.66 [dc] | 6.37±0.20 [abc] | 20.17±0.69 [c] | 9.27±0.30 [a] | 29.44±0.43 [d] | 94.24±0.60 [e] |
| R_30min-160°C | 3.24 | 65.49±1.13 [c] | 30.26±0.52 [dc] | 5.99±0.47 [bc] | 26.52±0.51 [a] | 8.41±0.08 [bc] | 34.93±0.54 [ab] | 95.97±0.33 [ab] |
| R_30min-180°C | 3.83 | 61.13±1.47 [e] | 31.55±0.98 [abc] | 4.49±0.14 [d] | 26.90±0.79 [a] | 6.42±0.26 [d] | 33.33±0.89 [bc] | 95.28±0.18 [bcd] |
| R_45min-25°C | | 86.61±0.24 [a] | 29.61±0.27 [d] | 5.77±0.47 [c] | 20.07±0.30 [c] | 9.23±0.35 [a] | 29.29±0.42 [d] | 94.61±0.28 [de] |
| R_45min-140°C | 2.83 | 69.10±0.70 [b] | 26.86±0.60 [e] | 6.67±0.17 [ab] | 24.65±1.02 [b] | 8.98±0.08 [ab] | 33.64±1.09 [abc] | 94.87±0.45 [cde] |
| R_45min-160°C | 3.42 | 64.58±0.76 [cd] | 30.56±1.56 [bcd] | 5.98±0.47 [bc] | 26.54±1.14 [a] | 8.61±0.33 [ab] | 35.14±1.46 [a] | 96.06±0.26 [a] |
| R_45min-180°C | 4.01 | 59.96±1.12 [e] | 32.03±1.16 [ab] | 4.32±0.43 [d] | 27.36±0.32 [a] | 6.20±0.37 [d] | 33.56±0.18 [abc] | 95.55±0.27 [abc] |
| R_60min-25°C | | 86.61±0.90 [a] | 30.46±0.38 [bcd] | 5.71±0.61 [c] | 19.89±1.08 [c] | 9.34±0.31 [a] | 29.23±1.31 [d] | 94.74±0.59 [de] |
| R_60min-140°C | 2.96 | 68.56±1.21 [b] | 27.98±0.95 [e] | 6.93±0.64 [a] | 24.79±0.54 [b] | 8.68±0.26 [ab] | 33.47±0.29 [abc] | 95.66±0.50 [ab] |
| R_60min-160°C | 3.54 | 63.71±1.08 [d] | 30.61±1.88 [bcd] | 6.04±0.42 [bc] | 27.30±0.38 [a] | 7.76±0.71 [c] | 35.05±1.04 [ab] | 95.91±0.12 [ab] |
| R_60min-180°C | 4.13 | 59.56±1.23 [e] | 32.80±0.78 [a] | 4.17±0.32 [d] | 26.99±0.47 [a] | 5.75±0.26 [d] | 32.75±0.69 [c] | 94.91±0.17 [cde] |

[1] SF, severity factor; [2] SR, percentage solids recovery; [3] AIL, acid-insoluble lignin; [4] ASL, acid-soluble lignin; [5] TS, total solids content.

### 3.2. Second Stage

3.2.1. The Chemical Composition of Raw BW and Residue after HTE

The results of the composition of BW before and after HTE under different conditions are shown in Table 2. The contents of cellulose, hemicellulose, lignin, and total solid (TS) in raw BW were 27.77 ± 0.93%, 6.06 ± 0.18%, 26.12 ± 1.06% (acid-insoluble lignin 17.25 ± 0.73% and acid-soluble lignin 8.87 ± 0.73%), and 92.82 ± 0.39%, respectively. As the temperature and holding times increased from 25 °C to 180 °C and from 30 to 60 min, respectively, the percentage solid recovery decreased from 86.75 ± 0.85% to 59.56 ± 1.23%, indicating that the percentage solid recovery decreased as the SF increased. While the TS increased under different reaction conditions, the maximum of 96.06 ± 0.26% was achieved at HTE conditions of 160 °C for 45 min (SF = 3.42).

These results of the chemical composition analysis of the treated BW showed that the main component that degraded during HTE was hemicellulose. After HTE, the hemicellulose content decreased from 6.06 ± 0.18% to 4.17 ± 0.32%. Most hemicellulose and its degradation products converted to the liquid part after HTE. In contrast, for the raw BW, the contents of lignin (acid-soluble lignin and acid-insoluble lignin) in the residue increased from 26.12 ± 1.06% to 35.14 ± 1.46% as the SF increased under the different reaction conditions. The cellulose content also increased to 30% generally under the different reaction conditions, compared with 27.77 ± 0.93% in the control groups. Previous studies have reported that subcritical water can remove most hemicellulose and increase relative cellulose and lignin content in lignocellulosic materials such as corn stover, bamboo culm, and sugarcane straw [26,27,30]. The reasons may be as follows: first, hemicellulose is more easily decomposed at mild temperatures, followed by lignin, whereas cellulose is difficult to decompose at temperatures below 200 °C [31]. Second, the decrease in hemicellulose content and increase in TS may have also caused the relative contents of cellulose and lignin to increase. Third, BW contains a high content of antioxidants which can be transferred to the liquid phase by HTE, thereby increasing the relative contents of cellulose and lignin.

The percentage removal of cellulose, hemicellulose, and lignin of BW under different reaction conditions is shown in Figure 4. Compared with the control groups, the percentage removal of cellulose, hemicellulose, and lignin was clearly increased by HTE ($p < 0.05$), reaching maximum values of 33.17 ± 1.26%, 59.9 ± 1.06% and 25.33 ± 0.95%, respectively. Cellulose still remained in the residue after HTE, which might be caused by amorphous regions of cellulose being hydrolyzed under mild HTE conditions [32]. Therefore, more serious treatment (SF > 4.13) is necessary to remove the crystalline regions of cellulose. Compared with cellulose, lignin was more susceptible to degradation under different HTE conditions. The highest percentage removal was hemicellulose under HTE conditions. However, it is possible that hemicellulose was not completely removed due to the physical obstacle of the outer edges of the cell wall, where xylan can build up and combine with cellulose [33].

3.2.2. Inhibitory Effects of Enzymatic Hydrolysis of Residue after HTE

Enzymatic hydrolysis of the HTE residue produced a lower glucose yield than the control groups, as shown in Figure 5A–C. The lowest glucose yield was 3.71 ± 0.59% from residue produced at 180 °C for 60 min (SF = 4.13) in Figure 5C, but it did not change greatly with different SF values. The highest glucose yield of 7.61 ± 0.41% was obtained in the control group (25 °C for 30 min) (Figure 5A). These results were due to many factors affecting the enzymatic hydrolysis of lignocellulosic biomass, such as the porosity and particle size of the substrate, treatment conditions, enzymatic hydrolysis conditions, cellulose accessibility, lignin barrier, and hemicellulose content [34]. Recently, the lignin barrier has been given more research attention. The main reason for this is that nonproductive adsorption of lignin on the enzyme leads to the formation of lignin-enzyme complexes, which reduces the amount of enzymes available for cellulose degradation [35]. In the present study, the high proportion of lignin in the residue after HTE may be caused by the inhibition of enzymatic hydrolysis (Table 2). Shen et. al. reported similar findings, in which the klason lignin yield

at 180 °C was higher than the cellulose yield, with the saccharification lower than at 160 °C and 170 °C in enzymatic hydrolysis [36].

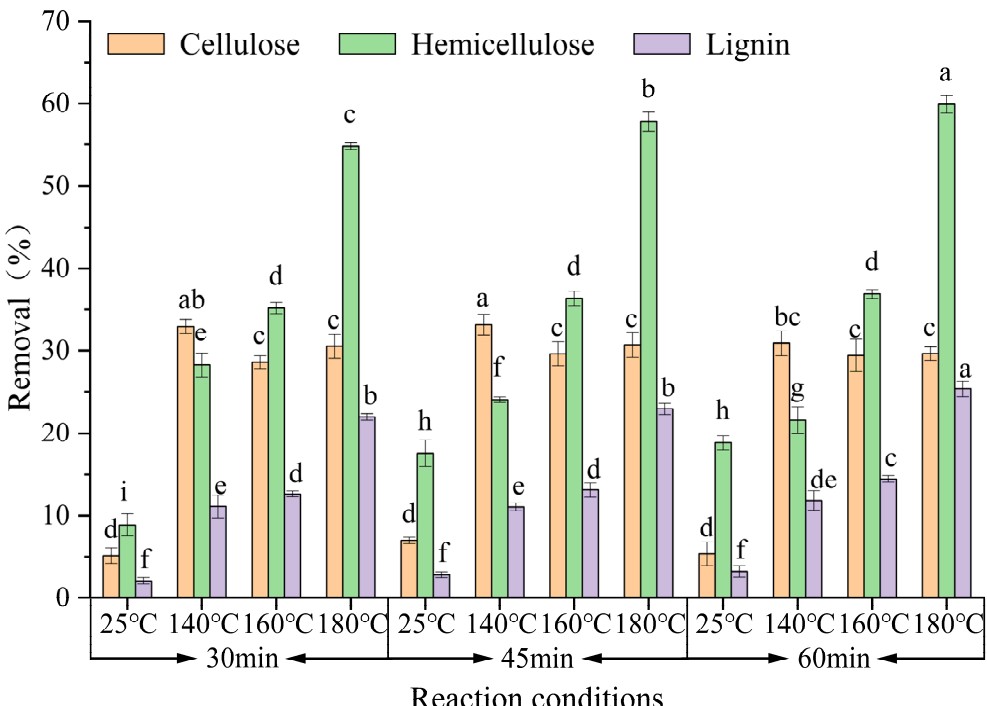

**Figure 4.** The effect of different HTE conditions on the percentage removal of lignocellulose from BW in the second stage. The alphabets on the bar graph indicate statistical difference (*p* < 0.05).

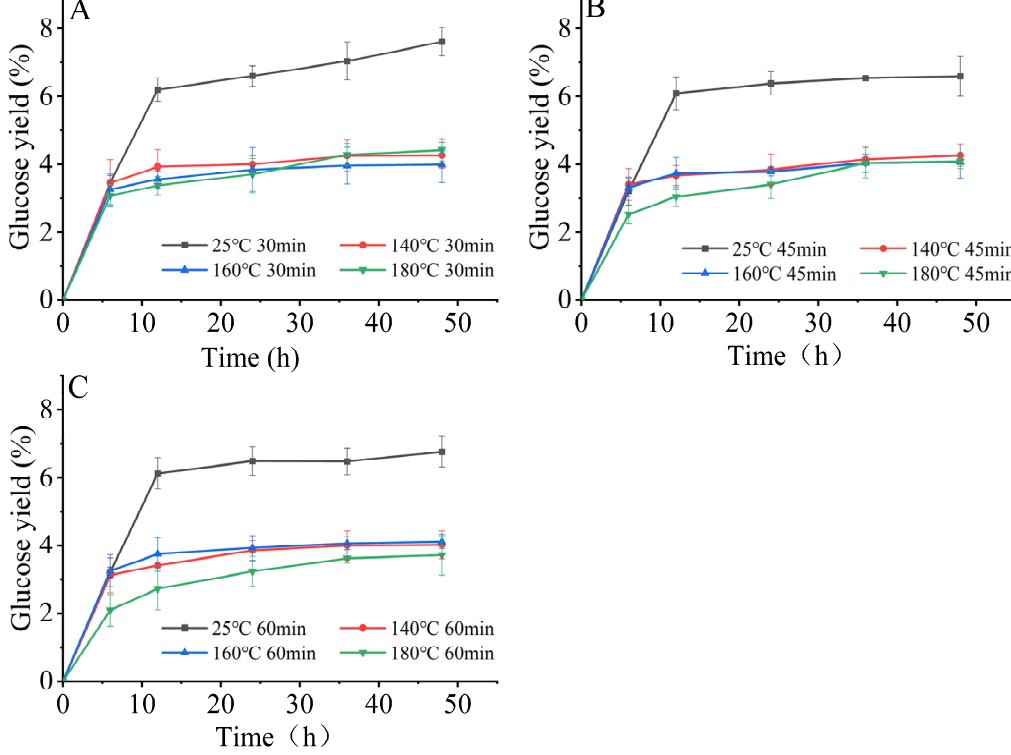

**Figure 5.** Effect of HTE conditions on the glucose yield after enzymatic hydrolysis for 48 h in the second stage. (**A**): 30 min with different temperature of THE, (**B**): 45 min with different temperature of THE, (**C**): 60 min with different temperature of THE).

Figure S2 shows the percentage degradation of the substrate after 48 h of enzymatic hydrolysis, illustrating that cellulose was degraded by cellulase to some degree. The lowest degradation was $26.24 \pm 1.24\%$ at 140°C for 60 min (SF = 2.96), with no large changes for different SF values ($p > 0.05$). The highest percentage degradation was $39.19 \pm 0.94\%$ in the control group (25 °C for 30 min). These data suggested that the percentage degradation in HTE residue was lower glucose production compared with those of the control group by enzymatic hydrolysis. Therefore, enzymatic hydrolysis is not advisable, as it would increase costs without achieving the desired effect of producing glucose.

### 3.2.3. Biochar Analysis after Pyrolysis

Biochar is a promising adsorbent for environmental contaminants. The physicochemical properties of biochar are closely related to the composition of the biowaste and the processing conditions [16,17,37]. After analysis of the composition of the residue from HTE and the results of the enzymatic hydrolysis experiments, during the second stage, the residue with its rich content of lignin was converted to biochar by pyrolysis, thereby reducing the difficulty and costs of degrading the HTE residue as well as fully utilizing the BW resources.

The decolorization of methyl orange (8 mg/L) and the biochar yield were investigated using raw BW and the residue after HTE treatment conditions of 160 °C for 30 min (SF = 3.24) (Figure 6). The HTE residue led to a slightly higher biochar yield of $34.45 \pm 0.78\%$ compared with that of raw BW of $33.68 \pm 0.70\%$ as shown in Figure 6. Gul et al. reported that the biochar from a lignin-rich sample could be of higher quality [38]. In the current study, the biochar production was not significantly different ($p > 0.05$). This may be due to the influence of pyrolysis parameters (temperature and time etc.) [39,40] and the optimization of the pyrolysis process should be considered in our next work. However, the raw BW and the residue exhibited significantly different ($p < 0.05$) decolorizations of methyl orange, of 21.92% and 31.11%, respectively, after 7 h of reaction. Figure S3 shows the VIS absorption spectra of methyl orange after 7h of reaction with raw BW, HTE residue, and methyl orange standard samples. The main absorption peak was at 464 nm. The biochar developed from the HTE residue of BW can adsorb methyl orange more easily than raw BW. SEM images (in Figure 7) were analyzed to explain this result. The surface of the raw BW was smooth (Figure 7A). After HTE, the irregular pores were observed due to the removal or depolymerization of some cellulose, hemicellulose, and lignin (Figure 7B). When converted to pyrolysis, the fibrillar structure was almost completely collapsed, and the sample morphologies of residue from 160 °C after 30 min became much rougher than those of raw BW (Figure 7 C,D). HTE plays a pretreatment role in the second-stage pyrolysis. It caused the surface of the residue at 160 °C after 30 min to be more easily broken in pyrolysis. Hence, the produced biochar of the HTE residue had a larger surface area compared with those of raw BW by pyrolysis. These findings implied that the increased surface areas facilitated the adsorption capacity of the dye.

### 3.2.4. Biomass Conversion

Figure 8 summarizes the steps of biomass conversion during the two-stage thermal-chemical process for the treatment of BW. Initially, 10 g of milled and screened BW consisting of 2.78 g of cellulose, 0.61 g of hemicellulose, and 2.61 g of lignin was treated by HTE at 160 °C for 30 min. Then the solids and liquid were separated to yield 6.55 g of residue consisting of 1.98 g of cellulose, 0.39 g of hemicellulose, and 2.29 g of lignin and liquid containing 0.41 g of mixture saccharides consisting of 0.02 g of cellobiose, 0.18 g of glucose, 0.18 g of xylose, and 0.03 g of arabinose, with 0.19 g of byproducts consisting of 0.08 g of acetic acid, 0.03 g of lactic acid, and 0.08 g of 5HMF. During the second stage, the HTE residue with its rich lignin content was heated at 400 °C for 30 min to modify the BW fibers and produce 2.26 g of biochar with its excellent adsorption properties. In this study, the two-stage thermal-chemical process from BW is efficient for production of saccharides and biochar. Yet, unextractable carbohydrates were still present in the solid phase. Moreover,

the biochar production and adsorption could still be improved. Therefore, future studies are required to improve the quantities and characteristics of the products to acquire a better technoeconomic evaluation of the whole process.

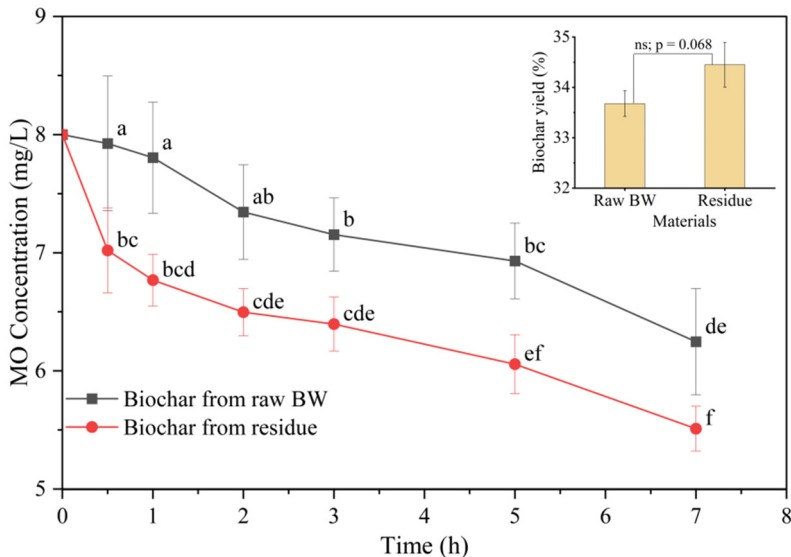

**Figure 6.** Biochar yield after pyrolysis in the second stage and decolorization of MO (methyl orange) during 7 h reaction by raw BW and residue of HTE (160 °C, 30 min). The alphabets on the bar graph indicate statistical difference ($p < 0.05$).

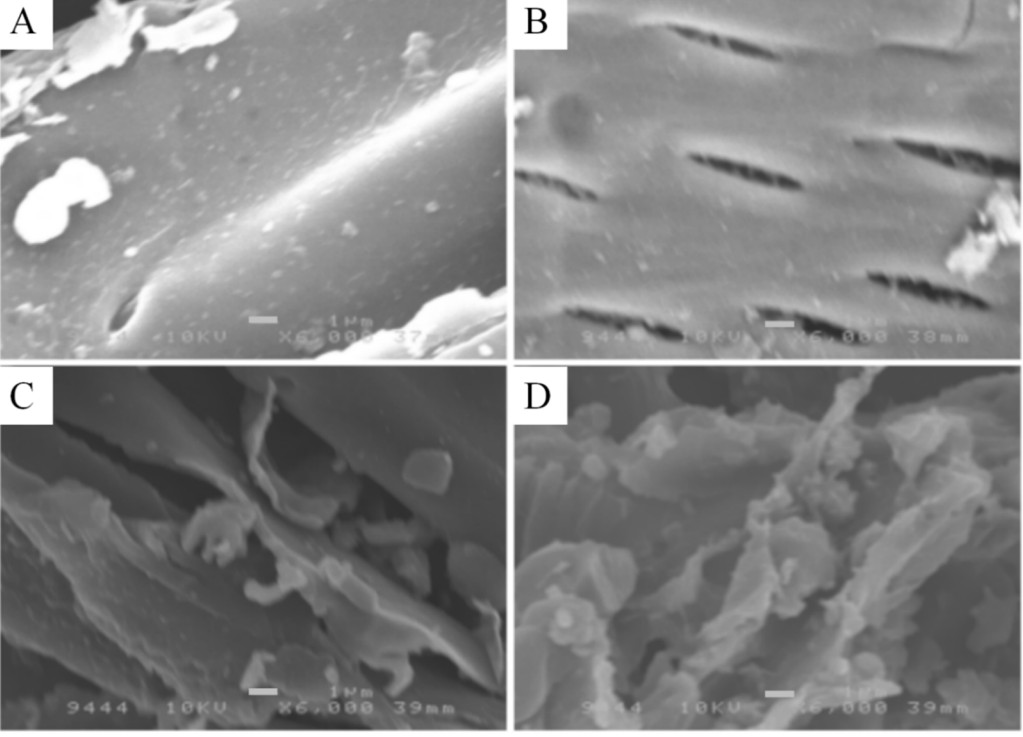

**Figure 7.** SEM images of biochar and control group: (**A**) Raw BW; (**B**) Residue of 160 °C, 30 min; (**C**) Biochar from raw BW; (**D**) Biochar from residue of 160 °C, 30 min.

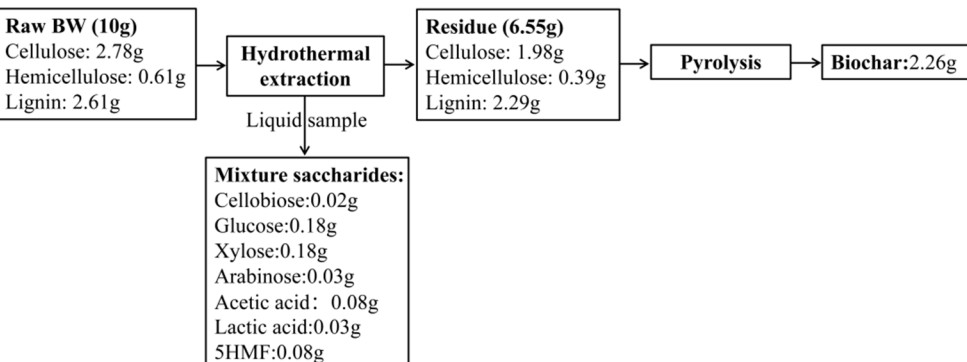

**Figure 8.** The evaluation of the biomass conversion of the two-stage thermal-chemical process.

## 4. Conclusions

In this study, the two-stage thermal-chemical process was explored to evaluate the production of saccharides and biochar. First, saccharides production from the buckwheat waste was feasible by HTE. Our study revealed that the SSY was 4.10%, with a lower total byproduct yield of 1.92% and a relatively higher percentage solid recovery of 65.49% under HTE conditions of 160 °C for 30 min (SF = 3.24). In addition, the adsorption of biochar from the residue was stronger than the biochar pyrolysis from raw BW. In particular, the decolorization of methyl orange was more pronounced to achieve 31.11% at 7 h. Therefore, our results indicated the feasibility of the two-stage thermal-chemical process combining HTE with pyrolysis to produce a substantial yield of saccharides and biochar from BW.

**Supplementary Materials:** The following supporting information can be downloaded at: https://www.mdpi.com/article/10.3390/fermentation8110573/s1, Figure S1: HPLC chart for glucan and xylan analysis of standard and liquid sample; Figure S2: Effect of HTE conditions on the percentage degradation of the substrate after enzymatic hydrolysis for 48 h. The alphabets on the bar graph indicate statistical difference ($p < 0.05$).; Figure S3: VIS absorption spectra of MO standard sample and raw BW as well as residue (160 °C, 30 min).

**Author Contributions:** Y.Y.: conceptualization, methodology, software, formal analysis, investigation, visualization, validation, data curation, writing—original draft preparation; F.L.: software, formal analysis, data curation, investigation, validation, writing—review & editing; N.H.: resources, validation, data curation, investigation; N.S.: resources, supervision, writing—review & editing, project administration, funding acquisition; B.Z.: formal analysis, investigation; Y.I.: formal analysis, investigation; R.N.: resources. All authors have read and agreed to the published version of the manuscript.

**Funding:** This study was funded by the Japan Science and Technology Agency, Support for Pioneering Research Intiated by the Next Generation (grant number JPMJSP2119). Yongheng Yuan is the recipient of a Hokkaido University Ambitious Doctoral Fellowship.

**Institutional Review Board Statement:** Not applicable.

**Informed Consent Statement:** Not applicable.

**Data Availability Statement:** The data in this paper are shown in the graphs in the paper.

**Acknowledgments:** The authors would like to thank Wada Tomonori (Hokkaido University) for providing the raw materials, and thank the Ibrahim Shaba Mohammed for the kind suggestion in my study.

**Conflicts of Interest:** The authors declare that they have no known competing financial interest or personal relationships that could have appeared to influence the work reported in this paper.

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
