# Peer review of "Exploring the Valorization of Buckwheat Waste: A Two-Stage Thermo-Chemical Process for the Production of Saccharides and Biochar"

_fermentation, doi:10.3390/fermentation8110573_

Round 1
Reviewer 1 Report
Following an overall inquiry into the reviewed article, I consider it to be a very interesting investigation of the valorization of buckwheat waste for production of saccharides and biochar.
The manuscript contains interesting data, which have been mostly correctly evaluated and interpreted. Organization and clarity of the manuscript is also generally good. The paper resolves an elaborate multidisciplinary topic and meets formal layout standards and default criteria, imposed on such articles.
Thereby, I recommend its issuance.
I would like to ask the authors to marginally mention the following article in the introduction: Viglašová E. et al.: (o) Production, characterization and adsorption studies of bamboo-based biochar/montmorillonite composite for nitrate removal. Waste Management 79: 385-394 (2018). (oo) Engineered biochar as a tool for nitrogen pollutants removal: Preparation, characterization and sorption study. Desalin. Water Treat. 191: 318-331 (2020).
Author Response
Response to Reviewer 1 Comments
Point 1: I would like to ask the authors to marginally mention the following article in the introduction: Viglašová E. et al.: (o) Production, characterization and adsorption studies of bamboo-based biochar/montmorillonite composite for nitrate removal. Waste Management 79: 385-394 (2018). (oo) Engineered biochar as a tool for nitrogen pollutants removal: Preparation, characterization and sorption study. Desalin. Water Treat. 191: 318-331 (2020).
Response 1: I have read these two papers carefully, and studies about the biochar and its composites on bamboo are very interesting and practical studies, because they provide an effective way to improve the adsorption properties of biochar and apply the adsorption properties of biochar to nitrogen adsorption, providing a novel avenue to use biochar and a valuable reference for our present and future research. Therefore we have cited this literature in our revised manuscript of introduction and results and discussion sections in Line 61-62 and 339 and added in No. 16 and No. 17 of reference section.

Reviewer 2 Report
This is good study conducted by the authors. If possible, then add the results of FTIR and XRD of the treated and untreated biomass. Also add 1-2 articles of 2022 to update the study.
Author Response
Response to Reviewer 2 Comments
Point 1: This is good study conducted by the authors. If possible, then add the results of FTIR and XRD of the treated and untreated biomass.
Response 1: Thanks to the kind suggestion of the reviewer, FTIR and XRD were not included in this study because these two machines are under repair in our school. Therefore, in this study, we detected SEM and explained the reasons for the result and its reaction mechanism by analyzing the surface morphology and cellulose hemicellulose and lignin, and we also compared the adsorption effect of carbon materials produced from residues via methyl orange adsorption experiments. In addition, FTIR and XRD will be implemented in future studies.
Point 2: Also add 1-2 articles of 2022 to update the study.
Response 2: We have included some update studies and cited these in revised manuscript of introduction and results and discussion sections in Line 35 and 350 and added in No. 1 and No. 39 and No. 40 of reference section.
